# Association between Nighttime Work and HbA1c Levels in South Korea

**DOI:** 10.3390/healthcare10101977

**Published:** 2022-10-09

**Authors:** Yeon-Suk Lee, Jae Hong Joo, Eun-Cheol Park

**Affiliations:** 1Graduate School of Public Health, Yonsei University, Seoul 03722, Korea; 2Department of Public Health, Graduate School, Yonsei University, Seoul 03722, Korea; 3Institute of Health Services Research, Yonsei University, Seoul 03722, Korea; 4Department of Preventive Medicine, Yonsei University College of Medicine, Seoul 03722, Korea

**Keywords:** shift work, work schedule, nighttime, HbA1c

## Abstract

Background: As the world has become a 24 h society, people’s demands have generated various work schedules, leading to an increase in workers’ health problems. The study aimed to investigate the association between nighttime work and HbA1c levels among South Korean adults over the age of 30. Methods: Participants were selected from the 2016–2019 Korea National Health and Nutrition Survey; those diagnosed with diabetes were excluded. The dependent variable was the HbA1c level reported in the KNHANES health examination report. The main independent variable was the participant’s work schedule. Work schedules were classified into three categories based on the participant’s report: (1) day; (2) night and overnight, and (3) other. Generalized multiple linear regression was used, and the significance level was defined as *p* < 0.05. Results: The participants comprised 4773 men and 4818 women. Those engaged in the “day” schedule served as the reference group. Among the male participants, the “night and overnight” group had significantly larger HbA1c (%) levels than the “day” group (β = 0.061, *p* = 0.0085). Among these nighttime male workers, HbA1c (%) levels were particularly higher in the people who were physically inactive (β = 0.094, *p* = 0.0031), slept less than 7 h (β = 0.108, *p* = 0.0009), and skipped meals (β = 0.064, *p* = 0.0401). Conclusion: Our results revealed an association of nighttime work with increased HbA1c levels in male participants. High-risk groups for HbA1c levels require careful observation of physical activity, sleeping time, and eating habits.

## 1. Introduction

Shift work has become an essential part of the modern society with the emergence of occupational conditions requiring 24 h labor in the fields of healthcare, manufacturing, wholesale and retail businesses [1]. The term “shift work” often refers to the employment practice designed to provide service across the entire day including nighttime and early morning. Recently, there has been a growing number of publications suggesting the association between shift work and negative health outcomes. In particular, work hours outside of the conventional daytime is likely to cause disruption of circadian rhythms, which could potentially lead to the development of chronic diseases, including metabolic disorders, cardiovascular disease, and cancer [2,3,4,5].

Diabetes is one of the most prevalent metabolic disorders worldwide. According to the International Diabetes Federation (IDF), 463 million people (9.3%) were suffering from it in 2019 [6,7]. Given the increased life expectancy due to the advances in medicine, population aged 65 years and older will account for 20% of the global population within the next few years [6]. Hence, this implies that the growing population of elderly people corresponds to the epidemiologic concept of increased prevalence of diabetes. The current trend estimates that a sharp increase in the prevalence of diabetes will result in as many as 700 million patients by 2045 [7]. In South Korea, diabetes is a substantial contributor to public health concerns, affecting nearly 10% of the national population, and has been reported as one of the leading causes of death among both sexes [6,7,8,9]. Thus, it is anticipated that diabetes will increase its burden on the healthcare system in the future.

Potential health-related risk factors associated with shift work, such as irregular living habits, work stress, and sleep complaints, can act as part of a diabetes outbreak [10,11,12]. It has been suggested that shift workers are more likely to experience the destruction of their biorhythms around the clock [13,14,15,16]. Of the various work schedules, those who conduct labor during the nighttime have reported lower scores in perceived health. Hemoglobin A1c (HbA1c) is a central biomarker for the presence and severity of hyperglycemia that is used as a predictive biomarker for diabetes [17]. Thus, this study aimed to investigate the potential effect of nighttime (i.e., night and overnight) work on HbA1c in comparison to the conventional daytime work.

## 2. Methods

### 2.1. Study Participants

We collected data from the 2016–2019 Korea National Health and Nutrition Examination Survey (KNHANES). The KNHANES is a self-reported nationally representative survey of South Koreans of all ages designed to gather annual national data on sociodemographic, economic, and health-related conditions and behaviors. Since 2007, the collected data have been subjected to an annual review and approval by the KCDCP Research Ethics Review Committee. The KNHANES comprises secondary data and is publicly available to researchers [18].

There were 32,379 participants in the 2016–2019 KNHANES. A total of 2435 people who had already been diagnosed with diabetes at the time of the survey were excluded to ensure the reliability of the results. Additionally, 9541 participants under the age of 30 were excluded because the majority of them did not undergo blood testing as part of the survey. We also excluded 10,812 participants with no response/other missing values from each variable’s survey items. Finally, a total of 9591 participants (4773 men and 4818 women) were selected for analysis.

### 2.2. Variables

HbA1c level, the dependent variable in this study, was measured in the blood samples collected after 8 h of fasting. HbA1c was considered as a continuous variable. Blood samples are among the key components of health examination reported in the KNHANES which was conducted at mobile examination centers that travel to the survey location [18].

The main independent variable was the work schedule. Work schedules were classified into three categories based on the survey report as follows: (1) day; (2) night and overnight; and (3) other (rotational, flexible, split, or irregular working hours/schedules).

Demographic, socioeconomic, health, and disease-related factors were also assessed to account for the confounding. The demographic factors included age (30~39, 40~49, 50~59, 60~69, and ≥70 years). The socioeconomic factors included educational level (≤elementary, middle, high, ≥college), occupational status (white, pink, blue), household income (low, middle, high), and household composition (one person, one-generation household, ≥two-generation household). The health-related factors included physical activity (active, inactive), smoking status (current smoker, former smoker, non-smoker), drinking status (2~4 times/week, 2~4 times/month, never or occasionally), sleeping hours (<7 h, ≥7 h), eating habits (eating three meals regularly, skipping meals), total energy intake (proteins, fats, carbohydrates; quintile 1, quintile 2, quintile 3, quintile 4, quintile 5), and body mass index (BMI)-defined obesity status (obese (≥ 25), normal (18.5~24.9), underweight (<18.5)). The disease-related factors included hypertension (hypertension (SBP ≥ 140 mm Hg), prehypertension (120 ≤ SBP ≤ 139 mm Hg), normal (SBP < 120 mm Hg)) and fasting glucose level (impaired fasting glucose (100~125 mg/dL), normal (<100 mg/dL)).

### 2.3. Statistical Analysis

The mean HbA1c level was calculated for each of the categorized variables included in the study. Analysis of variance (ANOVA) was performed to compare the mean HbA1c levels within each categorized variable to assess for significant differences. A generalized linear regression model (GLM) adjusted for confounding variables was used to assess the HbA1c level according to the reported work schedule. Those who reported the “day” schedule served as the reference group. The stratified, clustering, and weight variables developed by the KNHANES were applied to all the analyses to improve the representativeness of the sample and account for the limited proportion of participants retained in the final analysis. All the statistical analyses were performed using SAS version 9.4 (SAS Inc., Cary, NC, USA).

## 3. Results

Table 1 summarizes the general characteristics of the study population (male: 4773; female: 4818). Among the male participants, 4103 (86.0%) reported “day”, 302 (6.3%) reported “night and overnight”, and 368 (7.7%) reported “other” as their ordinary work schedule. Of the male workers, those who reported “night and overnight” had the highest mean HbA1c level (5.64) with the lowest standard deviation. The *p*-value for ANOVA within the work schedule was 0.0071.

Among the 4818 female participants, 4077 (84.6%) reported “day”, 581 (12.1%) reported “night and overnight”, and 160 (3.3%) reported “other” as their ordinary work schedule. Of the male workers, those who reported “night and overnight” had the highest mean HbA1c level (5.64) with the lowest standard deviation. The *p*-value for ANOVA was not statistically significant.

Table 2 summarizes the results of the generalized multiple regression analysis for HbA1c levels according to the types of work schedule. The HbA1c levels are represented as adjusted beta (β) coefficient in relation to “day” work schedules. Among the male workers, the “night and overnight” workers had the highest HbA1c levels (β = 0.061), and that was statistically significant (*p* = 0.0085). On the other hand, there was no significant difference among the female workers according to their reported work schedules.

Table 3 summarizes the subgroup results of generalized multiple linear regression for HbA1c levels stratified by physical activity, hours of sleep, and daily eating. Among the 302 male “night and overnight” workers, the average HbA1c level significantly increased to 0.108 (*p* = 0.009) for those who reported having less than 7 h of sleep daily, to 0.064 (*p* = 0.0401) for those who reported skipping meals, and to 0.108 (*p* = 0.0579) for those who intook the least amount of energy. Those who reported being physically inactive also showed an increased level of HbA1c (β = 0.094; *p* = 0.0031).

Figure 1 demonstrates the results of generalized multiple linear regression for HbA1c levels according to the specific work schedules. In the male participants, HbA1c levels increased the greatest among those who worked overnight (β = 0.092; *p* = 0.0247), followed by those who worked during the night (β = 0.500; *p* = 0.0441).

## 4. Discussion

After controlling for confounding variables such as demographic, socioeconomic, health, and disease-related factors, the male participants who worked during the night and overnight showed increased levels of HbA1c in comparison to those who worked during the conventional daytime. On the other hand, there was no significant difference in HbA1c levels among the female participants according to their ordinary work schedules. Sex differences in glycated hemoglobin levels have not been widely discussed [19]. Although no consensus has been reached regarding sex effects on HbA1c, hormonal changes during the menstrual cycle may account for the differences in HbA1c levels in male and female participants [19,20,21]. Previous literature suggested that women have a shorter red blood cell survival in comparison to men, and this may lower HbA1c levels. Other empirical evidence showed that estrogen plays a role in suppressing erythropoiesis in vitro and in vivo, which is implicated in lowering HbA1c levels in women [21].

Metabolism operates on circadian rhythms around the clock. Work schedules outside the standard daytime period may impair physiological activities and are likely to result in aberrant glucose homeostasis [22,23]. The findings of our study add empirical evidence to support the existing phenomenon that workers who work during the nighttime have increased HbA1c levels [23,24,25,26,27,28]. In particular, the study showed that HbA1c levels increased in the nighttime workers who were physically inactive, those who slept less than 7 h, and those who had the least amount of energy intake.

Exercise promotes glucose metabolism, which leads to elevated insulin sensitivity and lower blood sugar. Furthermore, lack of exercise can increase the risk of diabetes. Prior research publications showed that people with impaired fasting glucose disorder showed a statistically significant difference in the relative risk for non-exercising groups (1.375 for men and 1.124 for women), confirming the benefits of exercise [29]. Studies of diabetic and prediabetic men also showed that one to two physical activities a week were more effective in controlling blood sugar than inactivity [29,30].

HbA1c levels of people working during the nighttime with less than seven hours of sleep were significantly higher. Adequate sleeping time can help the body recover from fatigue and maintain biorhythms. According to the recommended sleeping time for each age group provided in the National Sleep Foundation (NSF) guidelines for 2015, the optimal sleeping time for adults is more than seven hours [31]. Structural changes in modern society are affecting the amount and quality of sleep as shifts increase and various types of working hours arise [32]. In addition, the amount and quality of this poor sleep are consistent with previous studies that linked them with the risk of type 2 diabetes [33,34].

Regarding eating habits, higher HbA1c levels are found in people who skip more than one meal a day compared to those who regularly eat three meals a day while working evenings and nights. Dining patterns have changed to Westernized eating habits since industrialization, which has also affected the development of diabetes, closely related to insulin secretion [30,31,32,33,34]. Carlson et al. assessed the effect of glucose metabolism on healthy men and women of normal weight without decreasing energy consumption [35,36]. They found that the morning fasting blood sugar level of the meal skipping group was higher than that of the group who regularly ate three meals a day. A prior study that investigated the relationship between the regularity of meals and impaired blood glucose disorder in adult non-diabetes groups also showed aligned results, with a 1.27 times higher incidence of impaired fasting glucose disorder in the groups that skipped meals [36,37].

In this study, HbA1c levels did not reach the prediabetes level. However, previous studies reported higher rates of the incidence of diabetes for five years according to glycated hemoglobin levels. Notably, HbA1c levels increased to less than 9% if they previously were 5.0–5.5%, to 9–25% if they previously were 6.0–6.5%, and to 25–50% if they previously were 6.0–6.5% [2]. Another previous study tracked HbA1c levels for six years, excluding those with a history of diabetes. It showed the highest sensitivity (59%) and specificity (77%) when the HbA1c level was initially 5.6%, with a 2.4 times increase in men and 3.1 times increase in women [38].

This study has several limitations. First, the cross-sectional design rendered us unable to determine a causal relationship between nighttime work and HbA1c levels. Second, the key covariates considered in this study, including physical activity, sleeping time, and eating habits were self-reported, possibly subject to recall bias. Third, the duration in working years of the current shift work was not considered. Lastly, the types of shift work were not specified in the survey data, and the workers were only specified by “collar colors”. Despite these limitations, this study also has certain strengths. First, it was the first study to analyze the association between shift work and HbA1c for non-diabetes groups in South Korea. Second, the data used in this study came from a national health statistics survey that calculated more than 500 health indicators such as eating habits and chronic diseases. The National Health and Nutrition Survey comprised a representative sample of the South Korean population, representing the actual health status of Koreans, and an updated questionnaire every year can identify changes in health status. Therefore, it can contribute to improving the health of the people by reflecting on health policies.

## 5. Conclusions

This study investigated the association between nighttime work and HbA1c levels among Korean adults aged 30 or older. We found a significant association between working night shifts and HbA1c levels in the men. Lack of exercise, sleeping less than 7 h, and skipping more than one meal a day were especially highly associated with increased HbA1c levels. High-risk groups with increased HbA1c levels require careful monitoring of physical activity, sleeping time, eating habits, and BMI. However, our study could not determine a causal relationship between HbA1c and night work, and further studies are needed for validation.

## Figures and Tables

**Figure 1 healthcare-10-01977-f001:**
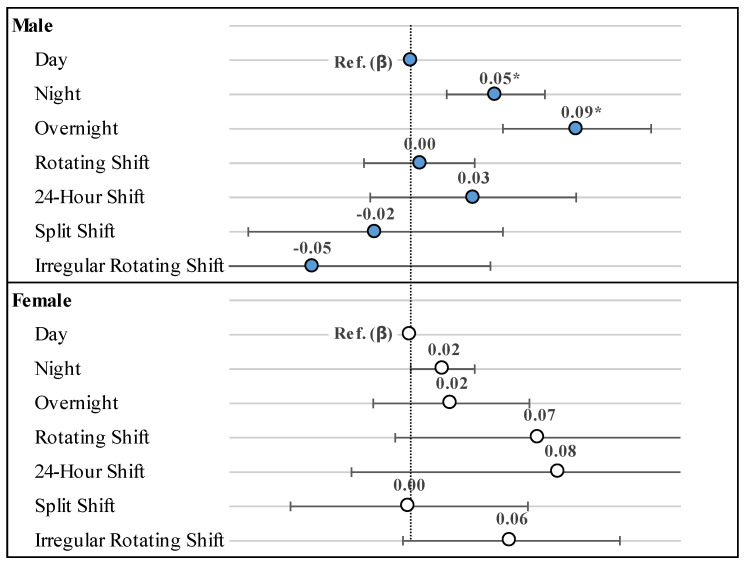
Generalized multiple linear regression for HbA1c levels according to the specific types of work schedules; * *p*-value < 0.05.

**Table 1 healthcare-10-01977-t001:** General characteristics of the study subjects and analysis of variance for the HbA1c levels.

Variables	HbA1c (%)													
	Total		Male						Female					
	*n*	%	*n*	%	Mean ± SD			*p*-Value	*n*	%	Mean ± SD			*p*-Value
**Total**	**9591**	**100.0**	**4773**	**49.8**	**5.60**	**±**	**0.39**		**4818**	**50.2**	**5.55**	**±**	**0.34**	
**Work schedule**								0.0071						0.2566
Day	8180	85.3	4103	86.0	5.59	±	0.43		4077	84.6	5.55	±	0.42	
Night and overnight	883	9.2	302	6.3	5.64	±	0.46		581	12.1	5.56	±	0.42	
Other	528	5.5	368	7.7	5.64	±	0.49		160	3.3	5.60	±	0.44	
**Age**								<0.0001						<0.0001
30–39	2107	22.0	1135	23.8	5.43	±	0.31		972	20.2	5.31	±	0.30	
40–49	2623	27.3	1253	26.3	5.53	±	0.42		1370	28.4	5.42	±	0.34	
50–59	2441	25.5	1138	23.8	5.65	±	0.41		1303	27.0	5.64	±	0.37	
60–69	1615	16.8	821	17.2	5.73	±	0.47		794	16.5	5.78	±	0.45	
≥70	805	8.4	426	8.9	5.83	±	0.58		379	7.9	5.90	±	0.51	
**Educational level**								0.8381						0.8638
≤Elementary	1404	14.6	515	10.8	5.74	±	0.49		889	18.5	5.79	±	0.50	
Middle	919	9.6	446	9.3	5.72	±	0.48		473	9.8	5.68	±	0.41	
High	2982	31.1	1418	29.7	5.62	±	0.46		1564	32.5	5.54	±	0.40	
≥College	4286	44.7	2394	50.2	5.53	±	0.39		1892	39.3	5.43	±	0.34	
**Occupational status**								0.0719						0.0607
White	3936	41.0	1952	40.9	5.52	±	0.39		1984	41.2	5.43	±	0.35	
Pink	1886	19.7	565	11.8	5.61	±	0.44		1321	27.4	5.60	±	0.45	
Blue	3769	39.3	2256	47.3	5.66	±	0.47		1513	31.4	5.68	±	0.44	
**Household income**								0.451						0.3665
Low	1022	10.7	386	8.1	5.72	±	0.58		636	13.2	5.74	±	0.50	
Middle	5143	53.6	2,621	54.9	5.60	±	0.42		2522	52.3	5.56	±	0.42	
High	3426	35.7	1,766	37.0	5.56	±	0.43		1660	34.5	5.48	±	0.37	
**Household composition**								0.5193						0.3746
One-person household	936	9.8	412	8.6	5.56	±	0.47		524	10.9	5.70	±	0.53	
One-generation household	2280	23.8	1238	25.9	5.68	±	0.49		1042	21.6	5.66	±	0.44	
≥Two-generation household	6375	66.5	3123	65.4	5.57	±	0.41		3252	67.5	5.50	±	0.38	
**Physical activity**								0.7231						0.9600
Active	4086	42.6	2161	45.3	5.57	±	0.43		1,925	40.0	5.54	±	0.41	
Inactive	5505	57.4	2612	54.7	5.62	±	0.45		2,893	60.0	5.56	±	0.43	
**Smoking status**								<0.0001						0.4196
Current smoker	1864	19.4	1648	34.5	5.61	±	0.45		216	4.5	5.47	±	0.37	
Former smoker	2381	24.8	2100	44.0	5.62	±	0.45		281	5.8	5.43	±	0.45	
Non-smoker	5346	55.7	1025	21.5	5.54	±	0.40		4,321	89.7	5.57	±	0.42	
**Drinking status**								<0.0001						<0.0001
2–4 times/week	2471	25.8	1824	38.2	5.57	±	0.42		647	13.4	5.41	±	0.38	
2–4 times/month	2289	23.9	1264	26.5	5.58	±	0.45		1025	21.3	5.47	±	0.40	
Never or occasionally	4831	50.4	1685	35.3	5.64	±	0.44		3146	65.3	5.61	±	0.43	
**Hours of sleep**								0.0426						0.2398
<7 h	3779	39.4	1932	40.5	5.62	±	0.46		1847	38.3	5.59	±	0.41	
≥7 h	5812	60.6	2841	59.5	5.58	±	0.42		2971	61.7	5.53	±	0.43	
**Eating habits**								0.0975						<0.0001
Eating three meals regularly	5438	56.7	2773	58.1	5.64	±	0.46		2665	55.3	5.63	±	0.43	
Skip meal (s)	4153	43.3	2000	41.9	5.54	±	0.41		2153	44.7	5.46	±	0.39	
**Total energy intake (kcal) ^a^**								0.4732						0.8141
Quintile 1	1919	20.0	956	20.0	5.64	±	0.48		963	20.0	5.56	±	0.46	
Quintile 2	1918	20.0	954	20.0	5.62	±	0.49		964	20.0	5.57	±	0.40	
Quintile 3	1919	20.0	955	20.0	5.58	±	0.41		964	20.0	5.56	±	0.43	
Quintile 4	1918	20.0	954	20.0	5.60	±	0.44		964	20.0	5.55	±	0.43	
Quintile 5	1917	20.0	954	20.0	5.54	±	0.35		963	20.0	5.53	±	0.39	
**BMI (kg/m^2^) ^b^**								<0.0001						<0.0001
Obese (≥25)	3340	34.8	2015	42.2	5.66	±	0.44		1325	27.5	5.71	±	0.46	
Normal (18.5~24.9)	5964	62.2	2669	55.9	5.55	±	0.43		3295	68.4	5.51	±	0.39	
Underweight (<18.5)	287	3.0	89	1.9	5.51	±	0.35		198	4.1	5.33	±	0.34	
**Hypertension (mm Hg)**								0.0017						<0.0001
Hypertension (≥140)	2758	28.8	1582	33.1	5.71	±	0.47		1176	24.4	5.78	±	0.48	
Prehypertension (120~139)	2605	27.2	1520	31.8	5.58	±	0.43		1085	22.5	5.59	±	0.39	
Normal (<120)	4228	44.1	1671	35.0	5.51	±	0.39		2557	53.1	5.44	±	0.36	
**Fasting glucose level (mg/dL)**								<0.0001						<0.0001
Impaired fasting glucose (100~125)	3206	33.4	1930	40.4	5.79	±	0.49		1276	26.5	5.85	±	0.48	
Normal (<100)	6385	66.6	2843	59.6	5.47	±	0.35		3542	73.5	5.45	±	0.34	
**Year**								<0.0001						<0.0001
2016	2258	23.5	1144	24.0	5.57	±	0.45		1114	23.1	5.50	±	0.40	
2017	2382	24.8	1181	24.7	5.55	±	0.40		1201	24.9	5.53	±	0.41	
2018	2492	26.0	1220	25.6	5.58	±	0.41		1272	26.4	5.55	±	0.43	
2019	2459	25.6	1228	25.7	5.68	±	0.49		1231	25.6	5.64	±	0.43	

^a^ Total energy intake = (carbohydrates (g) × 4 kcal/g) + (proteins (g) × 4 kcal/g) + (fats (g) × 9 kcal/g). ^b^ Obesity status defined by body mass index (BMI) based on the 2014 Clinical Practice Guidelines for Overweight and Obesity in Korea.

**Table 2 healthcare-10-01977-t002:** Generalized multiple linear regression for HbA1c levels according to work schedule.

Variable	HbA1c (%)					
	Male			Female		
	β	SE	*p*-Value	β	SE	*p*-Value
**Work schedule**						
Day	Ref.			Ref.		
Night and overnight	0.061	0.023	0.0085	0.019	0.017	0.2687
Other	0.008	0.025	0.7557	0.049	0.032	0.1273
**Age**						
30–39	Ref.			Ref.		
40–49	0.056	0.015	0.0002	0.062	0.013	<0.0001
50–59	0.135	0.017	<0.0001	0.200	0.017	<0.0001
60–69	0.207	0.022	<0.0001	0.272	0.024	<0.0001
≥70	0.296	0.039	<0.0001	0.326	0.038	<0.0001
**Educational level**						
≤Elementary	−0.017	0.028	0.5471	0.009	0.026	0.7357
Middle	−0.020	0.026	0.4381	0.014	0.024	0.5605
High	−0.012	0.015	0.4321	−0.003	0.013	0.8335
≥College	Ref.			Ref.		
**Occupational status**						
White	Ref.			Ref.		
Pink	0.040	0.020	0.0460	0.014	0.014	0.3233
Blue	0.036	0.015	0.0166	−0.010	0.017	0.5349
**Household income**						
Low	0.005	0.031	0.8590	−0.001	0.022	0.9491
Middle	0.013	0.013	0.3162	0.013	0.011	0.2480
High	Ref.			Ref.		
**Household composition**						
One-person household	−0.040	0.021	0.0522	0.032	0.021	0.1370
One-generation household	−0.002	0.016	0.9219	0.010	0.015	0.4869
≥Two-generation household	Ref.			Ref.		
**Physical activity**						
Active	Ref.			Ref.		
Inactive	0.006	0.012	0.5834	−0.002	0.011	0.8164
**Smoking status**						
Current smoker	0.080	0.016	< 0.0001	−0.010	0.023	0.6642
Former smoker	0.015	0.015	0.3137	−0.027	0.023	0.2503
Non-smoker	Ref.			Ref.		
**Drinking status**						
2–4 times/week	−0.113	0.014	< 0.0001	−0.118	0.016	<0.0001
2–4 times/month	−0.046	0.015	0.0020	−0.035	0.013	0.0077
Never or occasionally	Ref.			Ref.		
**Hours of sleep**						
<7 h	0.021	0.012	0.0727	0.023	0.011	0.0393
≥7 h	Ref.			Ref.		
**Eating habits**						
Eating three meals regularly	Ref.			Ref.		
Skipping meals	−0.020	0.013	0.1198	−0.045	0.011	<0.0001
**Total energy intake (kcal) ^a^**						
Quintile 1	0.004	0.018	0.8165	−0.014	0.018	0.4402
Quintile 2	0.022	0.019	0.2519	0.000	0.016	0.9795
Quintile 3	Ref.			Ref.		
Quintile 4	0.018	0.018	0.3109	−0.007	0.016	0.6509
Quintile 5	0.000	0.017	0.9834	0.007	0.016	0.6596
**BMI (kg/m^2^) ^b^**						
Obese (≥25)	0.071	0.012	<0.0001	0.080	0.013	<0.0001
Normal (18.5~24.9)	Ref.			Ref.		
Underweight (<18.5)	−0.031	0.031	0.3094	−0.067	0.025	0.0079
**Hypertension (mm Hg)**						
Hypertension (≥140)	0.064	0.016	<0.0001	0.079	0.018	<0.0001
Prehypertension (120~139)	0.022	0.014	0.1189	0.030	0.014	0.0272
Normal (<120)	Ref.			Ref.		
**Fasting glucose level (mg/dL)**						
Impaired fasting glucose (100~125)	0.267	0.013	<0.0001	0.297	0.015	<0.0001
Normal (<100)	Ref.			Ref.		
**Year**						
2016	Ref.			Ref.		
2017	−0.021	0.016	0.1844	0.033	0.015	0.0321
2018	−0.003	0.017	0.8475	0.048	0.015	0.0018
2019	0.095	0.018	<0.0001	0.120	0.015	<0.0001

BMI—body mass index. ^a^ Total energy intake = (carbohydrates (g) × 4 kcal/g) + (proteins (g) × 4 kcal/g) + (fats (g) × 9 kcal/g). ^b^ Obesity status defined by BMI based on the 2014 Clinical Practice Guidelines for Overweight and Obesity in Korea.

**Table 3 healthcare-10-01977-t003:** Generalized multiple linear regression for HbA1c levels stratified by physical activity, hours of sleep, and daily eating.

Variable	HbA1c (%)												
	Day	Male (*n* = 4773)						Female (*n* = 4818)					
		Night and Overnight (*n* = 302)			Other (*n* = 368)			Night and Overnight (*n* = 581)			Other (*n* = 160)		
	Β	β	SE	*p*-Value	β	SE	*p*-Value	β	SE	*p*-Value	β	SE	*p*-Value
**Physical activity**													
Active	Ref.	0.023	0.034	0.4995	0.014	0.035	0.6842	−0.011	0.027	0.6791	−0.015	0.038	0.7016
Inactive	Ref.	0.094	0.032	0.0031	−0.002	0.037	0.9605	0.037	0.022	0.0899	0.092	0.048	0.0561
**Hours of sleep**													
<7 h	Ref.	0.108	0.033	0.0009	−0.049	0.038	0.2011	0.024	0.026	0.3685	0.053	0.048	0.2747
≥7 h	Ref.	0.034	0.032	0.2894	0.065	0.032	0.0434	0.014	0.022	0.5105	0.042	0.041	0.3095
**Eating habits**													
Eating three meals regularly	Ref.	0.053	0.033	0.1095	−0.008	0.035	0.8227	−0.006	0.025	0.8017	0.056	0.050	0.2661
Skipping meals	Ref.	0.064	0.031	0.0401	0.028	0.036	0.4494	0.035	0.022	0.1206	0.055	0.041	0.1852
**Total energy intake (kcal) ^a^**													
Quintile 1	Ref.	0.108	0.057	0.0579	0.067	0.077	0.3870	0.025	0.038	0.5129	0.132	0.110	0.2336
Quintile 2	Ref.	0.059	0.056	0.2922	−0.042	0.045	0.3539	0.008	0.038	0.8297	0.079	0.074	0.2863
Quintile 3	Ref.	0.057	0.041	0.1679	−0.075	0.052	0.1555	0.010	0.044	0.8155	0.036	0.052	0.4823
Quintile 4	Ref.	−0.021	0.050	0.6658	0.021	0.057	0.7097	0.021	0.036	0.5720	−0.011	0.057	0.8546
Quintile 5	Ref.	0.045	0.046	0.3293	0.070	0.053	0.1913	0.066	0.030	0.0298	0.070	0.047	0.1352

^a^ Total energy intake = (carbohydrates (g) × 4 kcal/g) + (proteins (g) × 4 kcal/g) + (fats (g) × 9 kcal/g).

## Data Availability

The data is publicly accessible on the website of KNHANES administered by the Korea Disease Control and Prevention Agency (https://knhanes.cdc.go.kr/knhanes/index.do, accessed on 4 October 2022).

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
