# Peer review of "Association between Nighttime Work and HbA1c Levels in South Korea"

_healthcare, 2022, doi:10.3390/healthcare10101977_

Round 1
Reviewer 1 Report
Title:
Your title did not adequately represent your methods and findings. In abstract you mentioned “The dependent variable was the 15 HbA1c level measured after at least eight hours of fasting. The main independent variable was the 16 patients’ working hours. Working hours were classified into three different times: daytime (6 am–6 17 pm), evening and nighttime (2 pm–8 am the following day), and rotating time (i.e., alternating 18 shifts) … Those engaged in daytime work served 20 as the reference group. Among male participants, evening and nighttime workers had significantly 21 higher HbA1c (%) levels than daytime workers (êžµ=0.061, P=0.0085). Please update your title accordingly.
Background:
Please clarify your background section further. Currently, in most cases it is not clear whether are you giving global or local evidence. Since you are providing both global and national evidence, it is required that you let the readers know which one is global which one is national.
Please define shift work. What did you mean by shift work? To justify the need of this study, please discuss biological plausibility- how and why working hours affect HbA1c level?
You mentioned there is association between shift work and HbA1c level? Please specify which shift?
Methods:
You did not collect data from another survey. This is called secondary data analysis. Please update your methods accordingly. You also need to cite the original survey.
Ethical issues: you have done a secondary analysis. Have you received permission or the data is open access? Please clarify in methods.
You mentioned “The confusion variables such as age, educational 110 level, occupational status, household income, household composition, physical activity, 111 smoking status, drinking status, sleeping time, eating habits, total energy intake, body 112 mass index, and hypertension, fasting glucose level were all adjusted.” These are not confusion variables but confounding variables. Please update.
Results: For Table 1, please specify the test you performed and high light all significant association in the text where you discussed the table.
Table 2 title is inadequate. It is good that you specified the analysis and dependent variable in the title. Please make sure your title reflects your dependent variables and purpose of this test as well. In text, do not provide details of any non-significant results. Instead, you may just mention that no evidence of statistically significant association was observed between ….. and …. And, only provide details of significant association.
Discussion:
Please cite for any information you provided other than your findings. For example, in paragraph one you provided some info without giving reference.
You might consider using the words ‘working shift’ instead of ‘shift work’.
Conclusion: Please strengthen your recommendation based on your findings.
Reviewer 2 Report
1. The section of the introduction should explore the literatures about the relationship between general characteristics and HbA1C.
2. Why the authors select these variables (general characteristics) to as independent variables?
3. About Table 2, why all variables were selected to as the independent variables to predict HbA1c?
4. The section of discussion should discuss the results about Table 1-3.
Reviewer 3 Report
Thank you for submitting your manuscript on the association between shift work and HgbA1c levels among adult workers in South Korea. This is an interesting study that uses data from a large national database to determine associations between shift work and HgbA1c levels in adults over 30.
Line 83, p. 2 - glycosylated hemoglobin referred to as "glycated" hemoglobin. Are they the same?
p. 3, lines 111-113 - variables referred to as "confusion" variables. Do you mean "confounding" variables?
Was the type of shift work studied? Did you have access to this data? (for example, health care worker, vs factory worker)
References are quite dated. Of the 34 references listed, only five are within the last five years.
What was the reason for making the age cut off 30?
Do you have any theories regarding why statistically significant HgbA1C levels were found in males but not females who worked evenings and nights? (lines 168-171 describe why diabetes occurs more often in persons with unhealthy lifestyle habits and the relationship between obesity and diabetes but not why men were more affected than women). Were the women in the study more likely to have higher BMI's?
I would recommend developing the section on implications of your findings for future research and for clinical practice. The main take-away message seems to be the need for more careful monitoring of these populations. Should additional research focus on interventions to reduce risks in these groups of shift workers?
Thank you again for submitting your work. I wish you every success as you continue your scholarly pursuits.
Author Response
Point 1: Line 83, p. 2 - glycosylated hemoglobin referred to as "glycated" hemoglobin. Are they the same?
Response 1:
I appreciate you for your comment. Glycosylated was a typo. It is now revised to glycated. They were meant to be the same.
Point 2: p. 3, lines 111-113 - variables referred to as "confusion" variables. Do you mean "confounding" variables?
Response 2:
“Confusion” is revised to “confounding”. It was misinterpreted during the translation.
Point 3: Was the type of shift work studied? Did you have access to this data? (for example, health care worker, vs factory worker)
Response 3:
Unfortunately, the type of shift work has not been surveyed in the KNHANES. I realize that this is one of the limation of the study. This was now included in the limiation section of the manuscript.
Point 4: References are quite dated. Of the 34 references listed, only five are within the last five years.
Response 4:
Thank you for your valuable comments.
We were trying to cite the articles with the most number of citations. The authors agreed that we need more newer articles. We added a few more citations.
Those with five within the last five years are the current estimates of the burden of the disease.
Point 5: What was the reason for making the age cut off 30?
Response 5:
The reason is the majority of those under the age of 30 years did not undergo the blood sample testing conducted by the KNHANES. Previously, this was not explained in the manuscript. I provided the explanation in the Study Participants section for the readers.
Point 5: Do you have any theories regarding why statistically significant HgbA1C levels were found in males but not females who worked evenings and nights? (lines 168-171 describe why diabetes occurs more often in persons with unhealthy lifestyle habits and the relationship between obesity and diabetes but not why men were more affected than women). Were the women in the study more likely to have higher BMI's?
Response 5:
We found a possible theory regarding the sex effects on HbA1c level. Hormonal changes during the menstrual cycle may account for this difference. The explanation in detail is now provided in the revised manuscript. Thank you for your meaningful comment.
Point 5: I would recommend developing the section on implications of your findings for future research and for clinical practice. The main take-away message seems to be the need for more careful monitoring of these populations. Should additional research focus on interventions to reduce risks in these groups of shift workers?
Response 5:
Your recommendation is absolutely right. It is true that the risk group needs more careful monitoring. The conclusion of the manuscript is revised accordingly.

Round 2
Reviewer 3 Report
Thank you for making revisions to clarify the manuscript. I am comfortable recommending it for publication.
Author Response
Comment 1:
Thank you for making revisions to clarify the manuscript. I am comfortable recommending it for publication.
Response 1:
Thank for very much for your time and concern.
It is much appreciated. I hope you to submit to MDPI again in the future.
Kind regards,
Jae Hong Joo